# Tactile sensory coding and learning with bio-inspired optoelectronic spiking afferent nerves

Hongwei Tan [1✉], Quanzheng Tao [2], Ishan Pande [1], Sayani Majumdar[1,3], Fu Liu [4], Yifan Zhou[1], Per O.Å. Persson[2], Johanna Rosen[2] & Sebastiaan van Dijken [1✉]

The integration and cooperation of mechanoreceptors, neurons and synapses in somato-sensory systems enable humans to efficiently sense and process tactile information. Inspired by biological somatosensory systems, we report an optoelectronic spiking afferent nerve with neural coding, perceptual learning and memorizing capabilities to mimic tactile sensing and processing. Our system senses pressure by MXene-based sensors, converts pressure information to light pulses by coupling light-emitting diodes to analog-to-digital circuits, then integrates light pulses using a synaptic photomemristor. With neural coding, our spiking nerve is capable of not only detecting simultaneous pressure inputs, but also recognizing Morse code, braille, and object movement. Furthermore, with dimensionality-reduced feature extraction and learning, our system can recognize and memorize handwritten alphabets and words, providing a promising approach towards e-skin, neurorobotics and human-machine interaction technologies.

[1] NanoSpin, Department of Applied Physics, Aalto University School of Science, P.O. Box 15100, FI-00076 Aalto, Finland. [2] Thin Film Physics, Department of Physics, Chemistry and Biology (IFM), Linköping University, SE-581 83 Linköping, Linköping, Sweden. [3] VTT Technical Research Centre of Finland Ltd., P.O. Box 1000, FI-02044 VTT Espoo, Finland. [4] Department of Electronics and Nanoengineering, Aalto University, P.O. Box 15500, FI-00076 Aalto, Finland. ✉email: hongwei.tan@aalto.fi; sebastiaan.van.dijken@aalto.fi

n biological somatosensory systems, sensing, transmitting, and processing of information rely on distributed and parallel networks of receptors, neurons, and synapses, which are compact and efficient for solving complex and unstructured real-world problems[1–3]. External stimuli with environment information are encoded to action potentials (spikes) that are transferred by neurons and synapses, and synergistically combined to process the detected information with neural coding and learning[1–3]. Inspired by somatosensory systems, neuromorphic devices have been developed to mimic biological spike-based sensing and processing with the aim to enhance their performance and to achieve smart functions, such as image recognition[4,5], visual information processing[6,7], speech recognition[8,9], smart sensing and sensorimotorics[10–13], and neuromorphic computing[14–16], etc, which are power consuming if realized in conventional computing architectures.

Emulation of tactile sensing and processing as humans do is important for future intelligent robotics and human–machine interactions[17]. Recently, bio-realistic spiking afferent nerves based on flexible organic electronics have been described wherein resistive pressure sensors, ring oscillators, and a synaptic transistor are combined to detect, convert, and integrate pressure information[18]. However, the non-plastic architecture limits the emulation and implementation of learning and memorizing capabilities that enable humans to learn from and adapt to their environment via touch. Bio-realistic mimicking of coding, processing, learning, and memorizing of tactile information via artificial spiking afferent nerves at the hardware level would greatly advance bio-inspired sensory systems through complex neural coding principles, but this has not been demonstrated yet.

Here, we report an optoelectronic spiking afferent nerve with sensing, neural coding, perceptual learning, and memorizing capabilities. Emulating the biological SA-I afferent nerve (Fig. 1a), our artificial system detects pressure information by multiple flexible MXene-based receptors, converts and codes detected

information to optical spikes by coupling light-emitting diodes (LEDs) to ring oscillators and edge detectors (functioning as special analog-to-digital converter, ADC)[10], and then integrates the coded optical spikes using optoelectronic synapses (OE synapse), which are synaptic photomemristors based on ITO/ZnO/NSTO (Fig. 1b). In our system, we use optical communications between distributed receptors and synaptic photomemristors because of the advantages of non-contact integration. This allows one photomemristor to process multiple sensory inputs via optical spikes, providing a simple emulation of the integration of multiple action potentials from various axon terminals of pre-neurons to dendrites of post-neurons via synapses. With the implementation of rate coding and temporal coding, which are two of the major biological neural coding principles, the optoelectronic spiking nerve is capable of not only detecting, combining and distinguishing simultaneous pressure inputs, but also recognizing Morse code, braille characters, and object movement. Moreover, with the realization of feature extraction (coding) and learning in a dimensionality-reduced architecture, our system is able to recognize and memorize handwritten alphabets and words.

## Results

**Bio-inspired optoelectronic spiking afferent nerve.** MXene, a two-dimensional metal carbide/nitride[19,20], is a promising candidate for flexible electronics. The lamellar structure of MXene shows distance between atomic layers and conductivity changes in response to external pressure and has been exploited for flexible pressure sensors[21–24]. In our system, we used $Ti_3C_2T_x$ MXene with high-crystalline quality (Supplementary Fig. 1) to fabricate flexible pressure sensors (Supplementary Fig. 2) with a wide-range pressure response up to 200 kPa (Fig. 2a, b and Supplementary Figs. 3 and 4), which fully covers the working range of biological receptors. We designed a LED-coupled ring oscillator and edge detector, working as an optical ADC (for

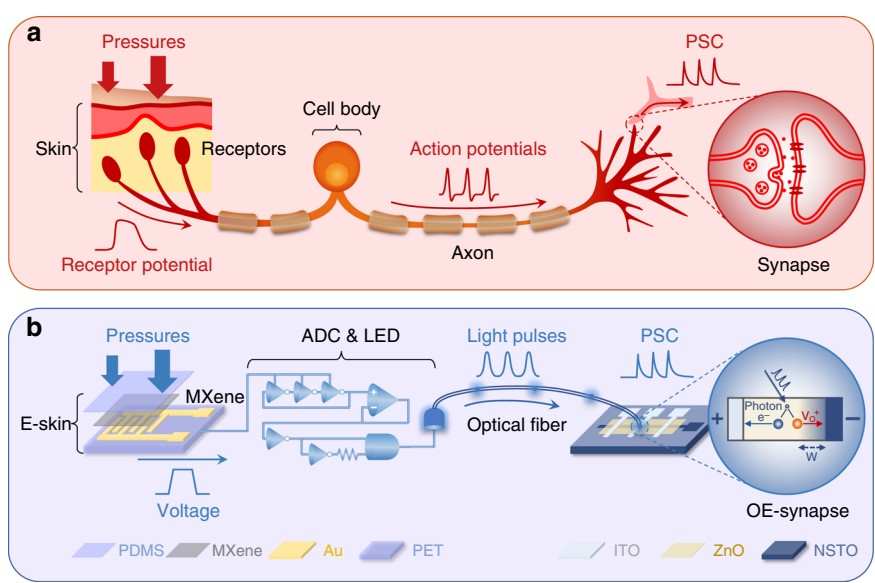

**Fig. 1 Schematic diagram of the biological and artificial afferent nerve systems. a** In the biological afferent nerve, external pressures applied to the skin change the potentials of receptors that are embedded in the skin. The cell body of the sensory neuron integrates the potentials and initiates action potentials (spikes) with coded pressure information. The axon transmits the action potentials to the axon terminals, which form synapses with interneurons, where they induce post-synaptic currents (PSCs). The central nervous system (CNS) processes the pressure information by integrating the action potentials from multiple synapses. **b** In the artificial afferent nerve, external pressures applied to the e-skin change the resistance of MXene in the flexible pressure sensor. The ADC-LED circuit, consisting of a ring oscillator, an edge detector and an LED, receives the voltage signal from the MXene sensor and initiates optical spikes with coded pressure information. The optical spikes are transmitted to a synaptic photomemristor (OE synapse), which integrates and processes the spikes into a PSC to decode and memorize the pressure information.

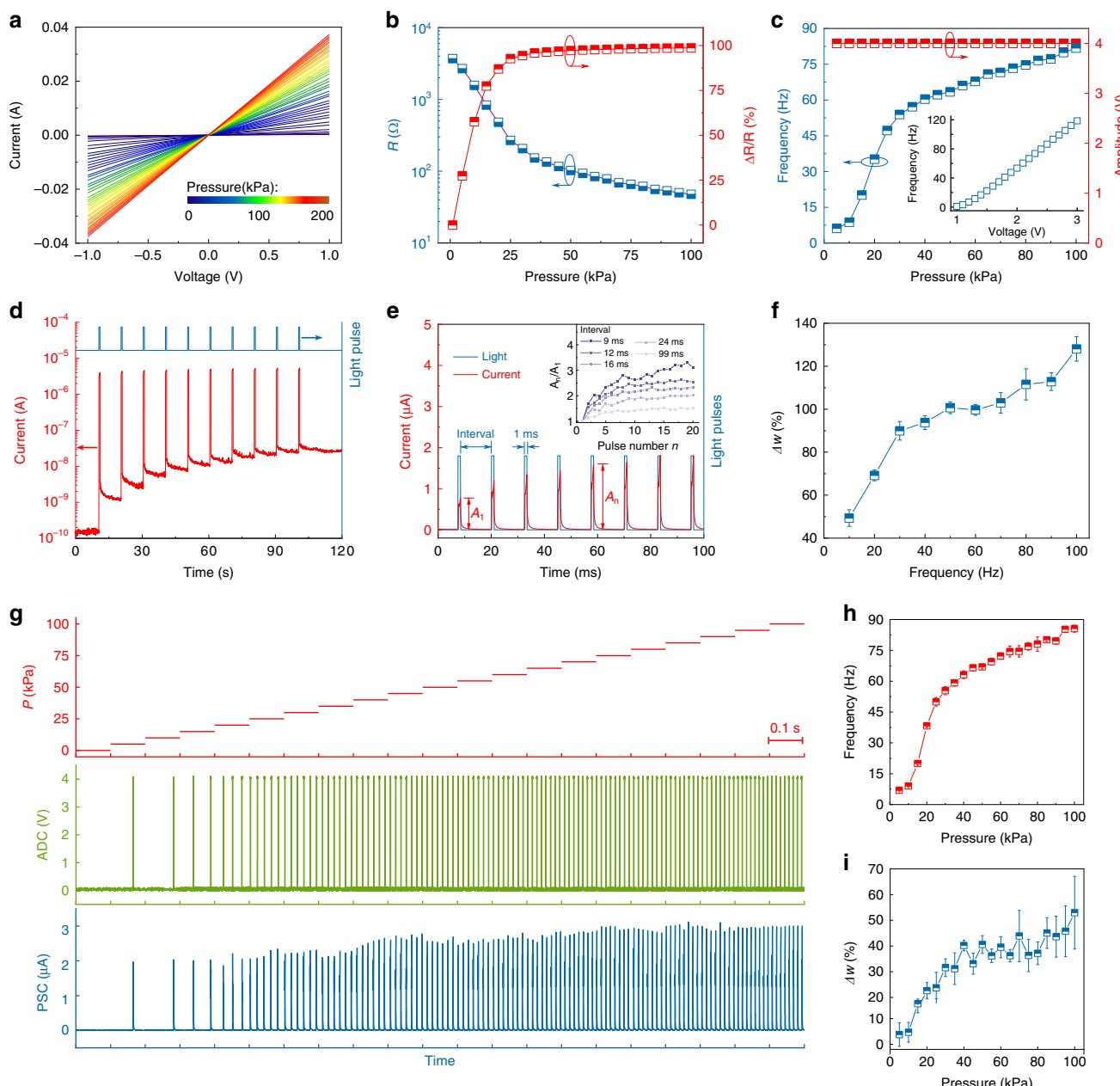

**Fig. 2 Characterization of the optoelectronic spiking afferent nerve. a** I–V curves of the MXene-based pressure sensor with applied pressures from 0 to 200 kPa. **b** Resistance and resistance change ratio in response to an increasing pressure. **c** Output frequency and amplitude of the pressure-dependent ADC for an increasing pressure up to 100 kPa. The insert shows the analog-to-digital conversion of electrical signals in the ADC. **d** I–t curve of the synaptic photomemristor with optical pulses as stimuli, showing current spikes and persistent photoconductivity (PPC) in response to the optical pulses. **e** Optical paired pulse facilitation (PPF) or neural facilitation behavior with respect to current spiking induced by optical spiking. **f** Spike-rate-dependent plasticity (SRDP) behavior with respect to PPC induced by optical spiking. **g** Input–output of the system showing the correlation among pressures, ADC outputs, and post-synaptic currents (PSCs). With increasing pressure, both the frequency of the PSC (**h**) and the weight change $\Delta w$ (**i**) increase. The error bars indicate variations during repeated measurements.

details on the electronic circuit see Supplementary Fig. 5), to convert pressure-dependent voltage signals to optical spikes (Fig. 2c). Bio-inspired spike coding is more robust than voltage amplitude coding because of voltage degradation and parasitic resistance issues in the latter coding scheme[18,25]. Moreover, the use of spike coding allows multiple coding principles, including rate coding, temporal coding, or a combination of both. Because of this, spike coding is capable of carrying larger volumes of data and distinguising multiple inputs with a single detector. To emulate the input–output of biological sensory neurons, the frequency range of optical output spikes was designed to be 0–100

Hz under a 0–100 kPa pressure input[18,26], while keeping the amplitude and duration of the optical spikes constant (Fig. 2c). In biological systems, action potentials carrying encoded information are conveyed to the central nervous system via synapses[1]. For our spiking afferent nerves, we designed a high-performance optoelectronic artificial synapse (Supplementary Table 1) using a synaptic photomemristor, which can not only generate a voltage spike, but also modify its weight (Fig. 2d and Supplementary Fig. 6) in response to optical spikes, thus allowing in-memory computation of sensory data. Moreover, it shows high sensitivity (Supplementary Fig. 7) and high working speed of up to 250 kHz

(Supplementary Fig. 8) that fully covers the working range of biological synapses. Owing to the persistent photoconductivity effect at the ZnO/NSTO interface (Fig. 2d), the optoelectronic synapse is capable of detecting and integrating the optical spikes into post-synaptic currents (PSCs) at the optical spike frequency (Fig. 2e and Supplementary Fig. 8). In addition, when stimulated by repeated optical spikes, the PSCs increase gradually and show neural facilitation behavior (Fig. 2e), which is essential for the transfer and processing of neural information[27,28]. Importantly, the synaptic weight changes at run-time by the input pressure because of a photomemristive effect. The weight change depends on the pressure amplitude, as illustrated by the spike-rate-dependent plasticity measured in response to 100 light pulses at different frequency (Fig. 2f). The input–output response of the full artificial spiking afferent nerve is shown in Fig. 2g. The relationships between the input pressure and the output frequency and weight change after applying a pressure for 1 s (Fig. 2h, i) demonstrate pressure-dependent spiking rate and weight modifications. For instance, an increase of input pressure to 100 kPa enhances the frequency of the PSCs to 86 Hz and produces a weight change of ~50%. The application of a short negative voltage pulse across the synaptic photomemristor quickly resets the PSC to its initial state (Supplementary Fig. 6), allowing the continuous execution of different tasks.

Utilizing the artificial spiking afferent nerve, we demonstrate a Morse code reader with temporal coding in Supplementary Figs. 9–11. In the temporal coding scheme, information is not only encoded in the timing and duration of spiking, but also in the timing and duration of non-spiking (quiescent)[29,30]. In our system, short and long spiking times ($t_{ss}$ and $t_{sl}$) in the PSC signal (Supplementary Figs. 9 and 10), resulting from short and long touching, correspond to dots and dashes in Morse code. The short and long non-spiking (quiescent) times ($t_{qs}$ and $t_{ql}$) indicate the spaces in and between Morse code letters, respectively. Values of the boundaries between time constants $t_{ss}$, $t_{sl}$, $t_{qs}$, and $t_{ql}$ can be trained and learned by statistically analyzing the PSC outputs induced by pressure inputs with Morse code information (Supplementary Fig. 9). As a demonstration, the name of our university 'AALTO' is recognized correctly (Supplementary Fig. 10). The flow chart of the Morse code recognition process and decoding program are shown in Supplementary Fig. 11 and Supplementary Note 1.

Besides temporal coding, Morse code could also be read using spike counting. This principle is widely used in biology and implemented easily. When Morse code characters are read, the optoelectronic memristor of the spiking afferent nerve produces a PSC signal comprising several groups of spikes (Supplementary Fig. 9). The total number of spikes is characteristic for each letter of the alphabet and thus could be used to read Morse code. For letters producing a similar amount of spikes, the recognition accuracy is improved by counting the spikes for certain groups (see insert of Supplementary Fig. 12 for 'R', 'U', and 'D').

As illustrated by the data in Fig. 2 and Supplementary Figs. 9–11, our optoelectronic spiking afferent nerve can utilize rate and temporal coding schemes. Rate coding returns the pressure amplitude during tactile sensing, which works well for a constant input. However, if the pressure fluctuates, any information contained in those fluctuations would show up as noise and, thus, be lost. Applications requiring information about the time evolution of pressure signals could successfully exploit a combination of rate and temporal coding, as demonstrated in Supplementary Fig. 13.

**Multiple integration and motion detection**. Leveraging the merits of non-contacting in optical communication and bio-inspired spike coding, a single synaptic photomemristor can combine and integrate multiple optical spike trains from different sensors and ADCs without complex electrical connections, providing a straightforward way of emulating the integration of action potentials from various axon terminals of pre-neurons to dendrites of post-neurons via synapses (Fig. 3a). As demonstrated by the data in Fig. 3b, the simultaneous application of 35 kPa (first panel) and 90 kPa (second panel) pressures to two sensors produces a PSC signal (third panel) that is comparable to the PSC sum of the individual measurements (fourth panel). After Fourier transformation, the frequency spectrum of the two-input spiking afferent nerve comprises two peaks corresponding to the 35 kPa and 90 kPa pressures (Fig. 3c). The ability to recognize simultaneous pressure inputs mimics the capability of SA-I afferent nerves in biological systems to distinguish different pressures and integrate action potentials with coded pressure information from multiple pre-neurons[18]. Using the functionality of handling multiple inputs, we demonstrate a braille reader in Supplementary Fig. 14. In the braille reader, a single synaptic photomemristor integrates the pressure-dependent optical spikes from two sensors when moving the sensors from top to bottom over a braille character. This produces a PSC, whose spiking rate and timing contain the braille information. With the braille dictionary of the alphabet and trained $F_0$, which separates the frequencies induced by touching the left and right convex patterns of a braille character, our system is capable of reading and recognizing braille, for example, the word 'HELLO', as shown in Supplementary Fig. 14.

Besides the detection and integration of multiple pressure information, our system is also capable of emulating the skin in registering the motion of objects by combining rate and temporal coding. We illustrate this function using a $2 \times 2$ sensor array wherein each of the sensing elements connects to an ADC-LED and a synaptic photomemristor (Fig. 3d). Whereas rate coding provides information about the pressure amplitude, the timing of spiking in the PSC signal of different optoelectronic synapses (Fig. 3e, f) indicates the direction of touching motion; left (sensor 1a) to right (sensor 2a). In addition, the spiking delay (latency to first spike) contains information about the touching speed. Using the physical distance between sensors 1a and 2a ($d_{2a-1a}$) and the latency time ($t_{2a}-t_{1a}$) between stimulus onset ($t_{1a}$) and first action potential ($t_{2a}$) in the PSC signal shown in Fig. 3e, the touching speed $v_1$ can be calculated as $v_1 = d_{2a-1a}/(t_{2a}-t_{1a})$. Similarly, other directions of touching motion can be detected too, as the results of Fig. 3g–j show. The detected touching speeds are summarized in Fig. 3k. We also fabricated a $4 \times 4$ sensor array (Fig. 3l and Supplementary Fig. 15) to advance the detection of touching motion in a larger area. From the PSC signals shown in Fig. 3m, the touching pressure (spiking frequency) and touching sequence are derived deterministically (Fig. 3n).

**Handwritten information processing**. When dealing with a large set of sensory data, feature extraction is widely used in machine learning to reduce the raw data to be informative and non-redundant, facilitating subsequent learning[31]. To simplify the processing of information in our system, we implemented feature extraction and feature learning in an architecture of multiple optoelectronic spiking afferent nerves (Supplementary Fig. 16) with reduced dimensionality[32]. In the design, every five sensors in a row of a $5 \times 5$ sensor array connect to an ADC-LED and a synaptic photomemristor, and we use this architecture to recognize handwriting through training (Fig. 4a). Spiking of a synaptic photomemristor indicates a touch in one of the five pressure sensors of a row. Instead of processing the 25 dimensional data stream from the $5 \times 5$ sensor array, we extract the spiking proportions of the five synaptic photomemristors as a five-dimensional (5D) feature for subsequent recognition and

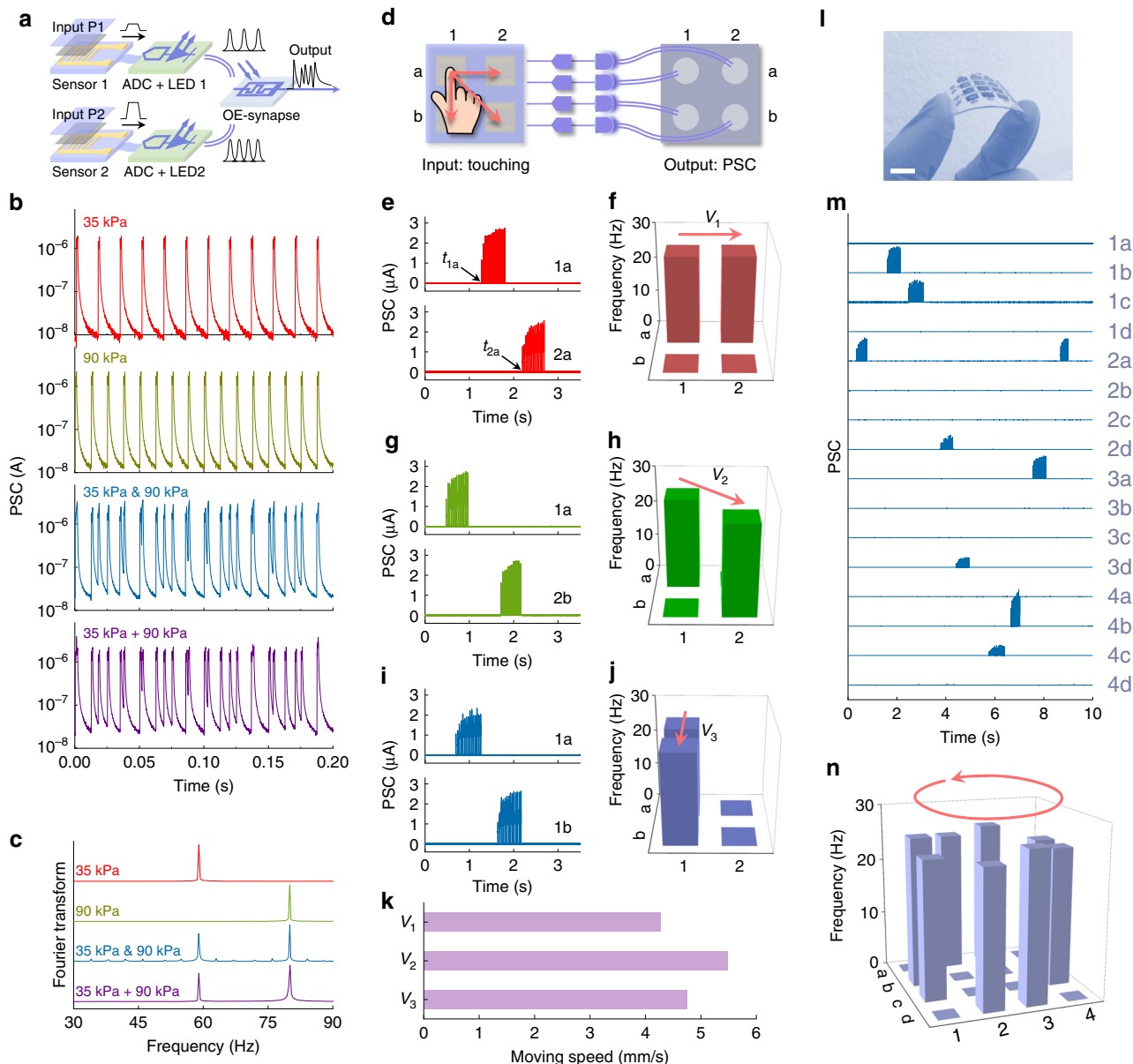

**Fig. 3 Pressures integration and motion detection. a** Schematic diagram of an optoelectronic spiking afferent nerve with two branches. **b** PSC measured with only one pressure 35 kPa (first panel) or 90 kPa (second panel) applied to one of the sensors. PSC measured with the same pressures applied to the two sensors simultaneously (third panel) and the numerical sum of the two PSCs shown in the first and second panel (fourth panel). **c** Fourier transformed spectra of the four signals shown in **b**. The peaks at 59 Hz and 80 Hz contain information on the pressure amplitude. One synaptic photomemristor can combine and integrate multiple pressures. **d** Schematic diagram of a 2 × 2 optoelectronic spiking afferent nerve for motion detection. **e, f** PSCs and frequencies detected when touching the sensor array from 1a to 2a. **g, h** PSCs and frequencies detected when moving a finger from sensor 1a to 2b. **i, j** PSCs and frequencies detected when touching the sensor array from 1a to 1b. **k** Average speed of touch motion in the three cases. **l** Image of a flexible 4 × 4 sensor array. The scale bar corresponds to 1 cm. **m** Detected PSCs from the sensor array when moving a finger over the array in circular motion. **n** Motion path and spiking frequency containing information on the pressure amplitude extracted from the PSCs in **m**.

learning processes (Fig. 4b). The spiking proportions ($P$) are defined as $P = t_{spiking}/t_{writing}$, where $t_{spiking}$ and $t_{writing}$ are the total spiking duration of a photomemristor and the time it takes to handwrite the letter (Fig. 4c). As an example, Fig. 4c shows the PSCs of the five photomemristors corresponding to the input of a handwritten 'A'. The five values of $P$, which are obtained directly after pressure input, form a 5D vector $\vec{P}$ (Fig. 4d). The insert of Fig. 4d shows $\overrightarrow{P_A}$ (the subscript indicates the letter) in a radar chart representation. In our architecture with reduced dimensionality, each written letter of the alphabet produces a different vector $\overrightarrow{P_l}$ ($l = A, B, C,…, Z$) (Supplementary Fig. 17). The 26

vectors form a complete alphabet dictionary of feature codes (Fig. 4e), which can be used for supervised and feature learning of handwritten inputs.

To demonstrate that the persistent photoconductivity of the five photomemristors in our 5D spiking afferent nerve facilitates learning, we used 10 sets of 26 vectors for training and another 10 sets for testing. During each training cycle, the vectors of the alphabet dictionary are updated by averaging the existing spiking proportions $\overrightarrow{P_l}$ ($l = A, B, C,…, Z$) and the new input $\overrightarrow{P_x}$ as ($\overrightarrow{P_l} + \overrightarrow{P_x}$)/2. The recognition process (Supplementary Note 2) evaluates the vector of a newly written letter by finding the best

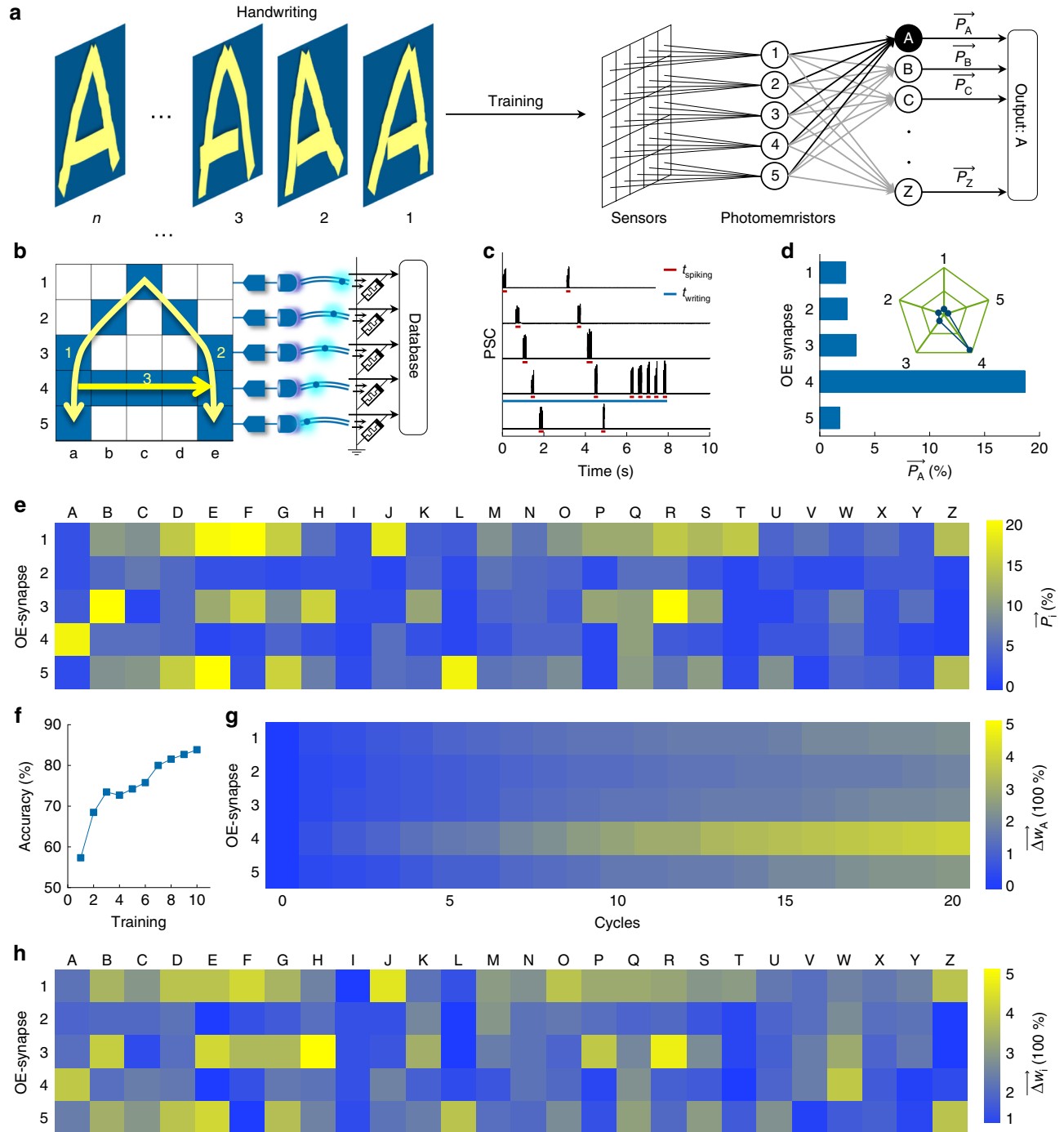

**Fig. 4 Recognition of handwritten letters of the alphabet with dimensionality-reduced features. a** Schematic diagram of handwriting recognition with feature extraction and feature learning using automatic hardware-based dimensionality reduction. **b** Structure and working principle of the optoelectronic spiking neural network. **c** Measured PSCs of five photomemristors with a handwritten 'A' as input. **d** Spiking proportions $\overrightarrow{P_A}$ extracted from **c**. The insert shows the relative activities of the five photomemristors with $\overrightarrow{P_A}$ in radar chart representation. **e** Learned feature dictionary of handwritten letters of the alphabet from the first input epoch. **f** Recognition accuracy of handwritten letters based on the learned feature dictionary. **g** Weight change evolution of the photomemristors during training and testing with a handwritten 'A' as input. **h** Weight change after 20 cycles for handwritten letters of the alphabet, demonstrating learning and memory capabilities.

matching vector in the dictionary (i.e., the smallest $|\overrightarrow{P_x} - \overrightarrow{P_l}|$ ($l = A, B, C,…, Z$)). After the first training cycle, the recognition accuracy is ~68% and it improves to 84% after 10 training cycles (Fig. 4f). The recognition error stems from variations in handwriting including writing stroke, speed, and path. Besides improving the recognition of $\overrightarrow{P_x}$, the weight of the OE synapse

(PSC of the photomemristor) also changes during the 10 learning and 10 testing cycles because of persistent photoconductivity in the ITO/ZnO/Nb-STO structure (Supplementary Fig. 5). Fig. 4g demonstrates that the weight change $\overrightarrow{\Delta w_A}$ during repeated writing of the letter 'A', where $\overrightarrow{\Delta w_A}$ is defined as (PSC$_2$-PSC$_1$)/PSC$_1$ and PSC$_1$ and PSC$_2$ are measured before and after stimulation, starts

to resemble $\overrightarrow{P_A}$ more and more during cycling. In fact, the complete color map of $\overrightarrow{\Delta w_l}$ ($l = A, B, C,\ldots, Z$) after 20 cycles shown in Fig. 4h is almost identical to the map of spiking proportions $\overrightarrow{P_l}$ in Fig. 4g. Feature conversion from spiking proportions $\overrightarrow{P_l}$ to memorized values of $\overrightarrow{\Delta w_l}$, as demonstrated further by their relationship in Supplementary Fig. 18, enables feature learning and memory of handwritten inputs.

Our dimensionality-reduced architecture with implemented feature extraction and feature learning provides a novel strategy for smart sensing and processing technologies. The extracted features $\overrightarrow{P_l}$ can be considered as a 'language' for human-machine or machine-machine communications. Compared with other systems (Supplementary Table 2), our optoelectronic spiking afferent nerve system demonstrates bio-realistic hierarchical architectures, optical spiking communication, and multiple coding principles. Moreover, the integration of optoelectronic memristors enables hardware-based dimensionality-reduced feature extraction and learning with recognizing and memorizing capabilities. A proof-of-concept demonstration of word recognition and memory on a letter-by-letter basis is shown in Supplementary Fig. 19. Here, the features ($\overrightarrow{P_l}$) and weight changes ($\overrightarrow{\Delta w_l}$) of 25 photomemristors are integrated to learn the handwritten word 'ESKIN' (i.e., five per letter), enabling the implementation of a 'bag-of-words' model[33].

Further dimensionality reduction in the classification of handwritten words is possible by combining the vectors of two subsequent letters. This attractive feature of our spiking afferent nerve system is demonstrated for the word 'APPLE' in Figs. 5a and 5b. In this realization, the dimensionality of the word is reduced from 25 to 15, limiting the number of required photomemristors. Similarly, the handwritten words 'ORANGE', 'BANANA', 'PEAR', 'CHERRY', and 'GRAPE' can be represented also by 15-dimensional vectors (Supplementary Fig. 20). To classify these words, we built an artificial neural network (Fig. 5c). The network consists of fifteen inputs and six outputs, corresponding to the elements of the 15-dimensional vectors and the six handwritten words, respectively. We trained the artificial neural network by repeated writing of the words. As shown in Fig. 5d–i, all words are recognized successfully after only four training cycles, as each output neuron responds to only one input word (Fig. 5j). In this proof-of-principle experiment, the words are relatively short. We therefore combined only two alphabet letters. For the processing of longer words, three or more alphabet letters could be combined to reduce the vector dimensionality further and recognize words efficiently.

## Discussion
Inspired by biological tactile sensing and processing in neural networks, we designed and demonstrated an artificial optoelectronic spiking afferent nerve with neural coding, perceptual learning, and

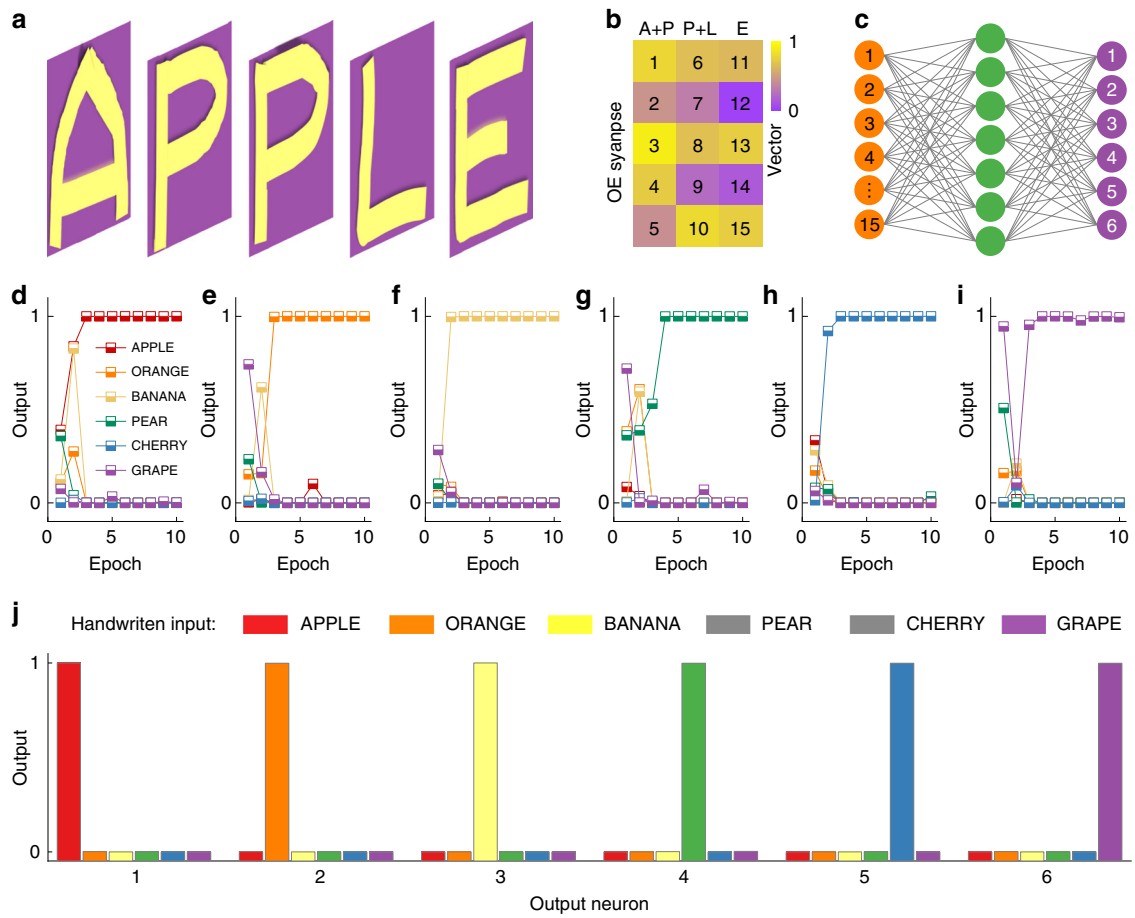

**Fig. 5 Classification of handwritten words with enhanced dimensionality reduction. a** Schematic diagram of the handwritten word 'APPLE'. **b** Dimensionality reduction of the vector representing 'APPLE' from 25 (five per letter) to 15 by combining 'A' and 'P', 'P', and 'L'. **c** Schematic diagram of the artificial neural network that processes the dimensionality-reduced vectors. The 15 vector elements are used as inputs and the six output neurons represent the six handwritten words. **d–i** Training results for the recognition of **d** 'APPLE', **e** 'ORANGE', **f** 'BANANA', **g** 'PEAR', **h** 'CHERRY', and **i** 'GRAPE' during the first 10 epochs. **j** Response of the six output neurons upon handwriting of the fruit names after the neural network is trained.

memorizing capabilities. Owing to its hierarchical structure and optical spike-based tactile sensing and processing features, our optoelectronic spiking afferent nerve recognizes Morse code, braille, and object movements. In addition, with the implementation of neural coding and perceptual learning based on activity-dependent spiking and weight modifications, our system can recognize, learn, and memorize handwritten letters and words, providing a promising strategy for artificial tactile sensation, computation in sensory memory, biomimetic sensors, smart optoelectronic prostheses, neurorobotics, and human–machine interactions.

## Methods

**2D MXene synthesis and characterization**. The $Ti_3AlC_2$ powder (parent material for the MXene, synthesized in house) and the MXene were investigated by means of X-ray diffraction and scanning electron microscopy combined with energy-dispersive X-ray spectroscopy, for compositional and structural analysis (not shown here). For more detailed structural analysis, high resolution transmission electron microscopy was performed in the double-corrected Linköping FEI Titan[3] 60–300, operated at 300 kV (Supplementary Fig. 1).

For MXene derivation, the etchant was prepared by adding 0.8 g of LiF to 10 mL of 9 M HCl and left under continuous stirring for 5 min. A total of 0.5 g of $Ti_3AlC_2$ powder (450 mesh) was gradually added (over the course of 5 min) to the etchant, and the reaction was allowed to run for 24 h at room temperature. The acidic mixture was washed with deionized $H_2O$ first via centrifugation (1 min per cycle at 4000 rpm) for two cycles. After each cycle, the acidic supernatant was decanted as waste followed by the addition of fresh deionized $H_2O$ before another centrifuging cycle. Then 3 M HCl and 1 M LiCl were used for additional washing via centrifugation (each for three cycles, 1 min per cycle at 4000 rpm). Finally, the mixture was washed with deionized $H_2O$ for another two cycles. These washing cycles were repeated until pH 4–5 was achieved. The final sediments were re-dispersed in deionized $H_2O$ (0.2 g MXene per 50 mL of water), deaerated with $N_2$, followed by sonication for 20 min. The mixture was then centrifuged for 30 min at 3000 rpm, and the supernatant was collected.

**MXene pressure sensors fabrication**. Commercial polyimide (PI) films were used as the flexible substrates of the sensors. Patterned Au/Ta electrodes with thickness of 50 nm/5 nm were deposited on PI substrates using sputtering (Ta: DC 30 W, Ar 30 sccm, 25 s. Au: DC 30 W, Ar 30 sccm, 300 s). PDMS films were used as capping layer. Before sticking to the patterned flexible substrate, plasma treatment (1 min) was used to make the surface of PDMS hydrophilic. Then, the MXene solution was dropped on the selected area and the solution evaporated in the air. Finally, the PDMS capping layer with MXene was aligned to the patterned area on the flexible PI substrate (Supplementary Fig. 2).

**Synaptic photomemristor fabrication**. Commercial conductive Nb-doped $SrTiO_3$ (NSTO) substrates were used as bottom electrode of the synaptic photomemristors. Photosensitive ZnO films with a thickness of 60 nm were grown by magnetron sputtering ($5.8 \times 10^{-3}$ mbar, Ar 16 sccm, O 4 sccm, power 60 W) on top of the NSTO substrates. This resulted in the formation of a Schottky barrier. Transparent and conductive ITO films grown by magnetron sputtering ($3.4 \times 10^{-3}$ mbar, Ar 10 sccm, power 50 W) through a metal shadow mask were used as top electrode. The working area of the synaptic photomemristors was 100 μm × 100 μm.

**Device and system characterization**. To test the pressure sensors, a force stand with integrated force gauge was used to apply pressures to the MXene-based sensors. The pressures were calculated according to the applied force and the area. I–V curves of sensors under different pressure loads shown in Fig. 2a were measured using a Keithley 4200 semiconductor characterization system. The electronic circuits (ring oscillator and edge detector) were tested using a Keithley 2400 sourcemeter and a Keysight DSO1024A oscilloscope. The synaptic photomemristors were measured using an Agilent B1500A semiconductor device parameter analyzer and 375 nm light pulses from an LED. The intensity of the light pulses was $0.65 \pm 0.06$ mW mm$^{-2}$, which was calibrated by a photodetector from Thorlabs (FD11A) and an optic spectrometer from Ocean Optics (USB2000+). To characterize the pressure-dependent PSC of the system, the force stand with force gauge was used to apply pressures and the Agilent B1500A was used to record the PSC. In measurements with multiple inputs, finger motion, or handwriting was performed on a sensor array and the PSCs of the synaptic photomemristors were recorded using the Agilent B1500A. The programs for data analysis and decoding were written using Wolfram Mathematica 12 and Matlab. 3D-printed blocks of braille characters were used to demonstrate braille recognition through pressure sensing.

## Data availability

The source data underlying the figures in the main manuscript and Supplementary Information are provided as Source Data file. All other data that support the findings of this study are available from the corresponding authors upon reasonable request.

## Code availability

The codes used in this study are included in the Supplementary Information or available from the corresponding authors upon reasonable request.

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

## Acknowledgements

We gratefully acknowledge N. Wei and E.I. Kauppinen for providing infrastructure support for the electrical measurements; S. Hiremath and T. Cui for fruitful discussion on feature learning; G. Liu and R.-W. Li for discussions on the photomemristor; C. Yu for substrates processing. This work was supported by the Academy of Finland (grant nos. 316973, 316857, and 13293916), and the Knut and Alice Wallenberg Foundation through a Fellowship grant and support to the Linköping Ultra Electron Microscopy Laboratory. The project made use of the Micronova Nanofabrication Center and the Aalto University Nanomicroscopy Center (Aalto-NMC), supported by Aalto University.

## Author contributions

H.T. and S.v.D. initiated the research. H.T. designed the systems. Q.T., J.R., and P.P. synthesized and characterized the MXene. H.T. and S.v.D. designed the synaptic photomemristors. H.T. fabricated the MXene-based sensors and photomemristors. H.T. and I.P. designed and prepared the circuits. H.T. and I.P. conducted the electrical and optoelectronic measurements. F.L. and Y.Z. wrote the program and code for data analysis and decoding. H.T., F.L., S.M., and S.v.D. analyzed the data. H.T. and S.v.D. wrote the manuscript with input from all the other authors. All authors discussed the results and commented on the manuscript.

## Competing interests

The authors declare no competing interests.
