## [Peer Review File · Nature Communications]

Reviewers' comments:

Reviewer #1 (Remarks to the Author):

In the manuscript titled "Tactile sensory coding and learning with bioinspired optoelectronic spiking afferent nerves", the authors reported the tactile sensor system with an optoelectronic artificial nerve. The sensory system has been made of MXene pressure sensor, LED-coupled with the analog-to-digital circuit, which converts electronic signal of the pressure sensor to optical signal and synaptic photomemristor based on. The authors demonstrated the tactile sensing system with recognition of Morse code, 5-letter-word, and braille character. The manuscript is concise and well-structured, and the optoelectronic tactile sensing system has originality, so the reviewer thinks that researchers in this field would be very interested in the contents of this manuscript. Therefore, the reviewer would recommend the publication of this paper to Nature Communications after the following minor revision. The following comments are not to criticize but to strengthen the manuscript.

1. Because the power of the light source can affect output current, the power of the light source need to be stated in the method section.
2. In Fig S9, PSCs among all Morse code letters do not seem different. So the reviewer thinks that Morse code recognition is mainly conducted by temporal coding, which is not a novel recognition method (needs additional processing unit). As a proof of concept in synaptic photomemristor, PSCs need to show bigger differences among letters as Lee, Y. Et al. Stretchable organic optoelectronic sensorimotor synapse. *Sci. Adv.* 4, eaat7387 (2018) does.
3. What is the difference or the originality of this study when compared with the previously reported system? The temporal coding seems not a novel approach that can be applied without a synaptic device. What is the most important point of this recognition system compared with other systems? This clarification could be critical for strengthening the originality and novelty of the research.
4. For artificial afferent nerve system, the retention time or decay time of PSCs are important. The biological sensor system and previously reported artificial afferent nerve showed a short retention time because the sensor signal needs to be decayed immediately right after the elimination of stimulation for accurate sensing. The retention time can be seen in Fig S6 or 2d, which is relatively long for an artificial sensory nerve system. The reviewer thinks that previously remained postsynaptic current may still affect the following results. So, can the recognition process be operated continuously with different patterns without sufficient time intervals?

Reviewer #2 (Remarks to the Author):

This manuscript is related on the emulation of tactile sensory coding and learning with artificial and bioinspired afferent nerves. The afferent nerves are made of a combination of devices including 2D material-based pressure sensors, ring oscillators with off-the-shelf electronics and photomemristors. Based on this system, functions related on tactile sensory and processing are demonstrated including processing of Morse code and braille alphabet. By having an array of pressure sensors and ring oscillators coupled with photomemristors, the authors also demonstrated dimensionality reduction of the data stream in hard-written letters. The manuscript is well organized and the results are presented in an easy to follow manner. However, the main argument of the manuscript is that in those systems a non-volatile end-device (memristor) is missing and this would enhance the functionality. I am not really sure about this argument for all demonstrations of the manuscript, since in the past Morse code and braille alphabet has been also demonstrated with volatile end-devices (*Science* 360, 6392, 998-1003 (2018) and *Science Advances* 4, 11, eaat7387 (2018)). Therefore, there is an overlap with the aforementioned works

and Figures 1-3 of the manuscript. Nevertheless, the actual merit of the manuscript as well as the advantage of having non-volatile end-devices is the demonstration of dimensionality reduction in hard-written letters (Figure 4). To conclude, I believe that this work presents some interesting concepts in dimensionality reduction and could be published in Nat Commun., if the manuscript is more situated around the concept of Figure 4.

Reviewer #3 (Remarks to the Author):

While the authors have developed an interesting intelligent mechanotransduction system, I don't see that this will be of general enough interest for Nature Communications, given the work of people like Gordon Cheng (A Comprehensive Realization of Robot Skin) or Zhu (Bilayered Oxide-Based Cognitive Memristor with Brain-Inspired Learning Activities). It seems more appropriate for a materials science or IEEE journal.

My concern for the most part isn't technical (the MXene-based sensor and photomemristor seems reasonable and technically well-conceived, with clever hardware-based processing; I also have no problems with the analysis used). It is more a matter of scope for a generalist journal. That said, I do wonder about the form factor of the different versions of the device?

Reviewer #1

Reviewer comment

In the manuscript titled “Tactile sensory coding and learning with bioinspired optoelectronic spiking afferent nerves”, the authors reported the tactile sensor system with an optoelectronic artificial nerve. The sensory system has been made of MXene pressure sensor, LED-coupled with the analog-to-digital circuit, which converts electronic signal of the pressure sensor to optical signal and synaptic photomemristor based on. The authors demonstrated the tactile sensing system with recognition of Morse code, 5-letter-word, and braille character. The manuscript is concise and well-structured, and the optoelectronic tactile sensing system has originality, so the reviewer thinks that researchers in this field would be very interested in the contents of this manuscript. Therefore, the reviewer would recommend the publication of this paper to Nature Communications after the following minor revision. The following comments are not to criticize but to strengthen the manuscript.

Our response

We would like to thank the reviewer for his/her positive assessment of our work and the recommendation to publish the paper in Nature Communications after minor revision. Below, we respond to the reviewer’s comments on a point-by-point basis.

Reviewer comment

Because the power of the light source can affect output current, the power of the light source need to be stated in the method section.

Our response

We calibrated the intensity of the LED light pulses using a photodetector from Thorlabs (FD11A) and an optic spectrometer from Ocean Optics (USB2000+). In the Methods section, we now state the measured value (0.65 ± 0.06 mW/mm²).

Reviewer comment

In Fig S9, PSCs among all Morse code letters do not seem different. So the reviewer thinks that Morse code recognition is mainly conducted by temporal coding, which is not a novel recognition method (needs additional processing unit). As a proof of concept in synaptic photomemristor, PSCs need to show bigger differences among letters as Lee, Y. Et al. Stretchable organic optoelectronic sensorimotor synapse. Sci. Adv. 4, eaat7387 (2018) does.

Our response

We thank the reviewer for this suggestion. In the original manuscript, we indeed employed temporal coding to recognize Morse characters. The use of amplitude, as demonstrated by Y. Lee et al. in *Sci. Adv.* 4, eaat7387 (2018) has the advantage of simplicity but also some disadvantages, including sensitivity to device-to-device variations, voltage degradation, and parasitic resistance issues. These factors limit the robustness of voltage amplitude coding and the differentiation between pressure inputs. Since our optoelectronic spiking afferent nerve system utilizes spikes, alternative coding schemes are possible. In the revised manuscript, we propose spike counting as a new robust mechanism for the recognition of Morse code. Spike counting is widely used in biology and implemented easily. It works as follows: When Morse code characters are read, the optoelectronic memristor of our spiking afferent nerve produces a PSC signal that comprises several groups of spikes (Figure S9). The total number of spikes is characteristic for each letter of the alphabet (see new Figure below). For letters producing a similar amount of spikes, the recognition accuracy is improved by counting the spikes of certain groups. In the example below, counting the number of spikes in group 1 and 2 provides clear differentiation between the letters 'R', 'U', and 'D'. We added the figure below to the Supplementary Information (Figure S12). A description of the spike counting mechanism is added to page 6 of the revised manuscript.

Fig. S12. Number of spikes produced by the optoelectronic spiking afferent nerve in response to pressure inputs from alphabet letters in Morse code. The output PSC signal of the optoelectronic memristor comprises several groups of spikes when Morse code characters are read (Figure S9). The total number of spikes defines each letter. If the spike count is similar (e.g. 'R', 'U', and 'D'), the number of spikes in the first and second spiking group could be used to enhance the recognition accuracy (see inserts).

Reviewer comment

What is the difference or the originality of this study when compared with the previously reported system? The temporal coding seems not a novel approach that can be applied without a synaptic device. What is the most important point of this recognition system compared with other systems? This clarification could be critical for strengthening the originality and novelty of the research.

Our response

Compared with previously reported spiking afferent nerve systems, especially the demonstrations in *Science Advances* **4**, eaat7387 (2018) (Ref. 13) and *Science* **360**, 998-1003 (2018) (Ref. 18), the novelty of our work can be summarized as follows:

1. With respect to the architecture, our system uses **optical communication** between discrete modules, allowing **non-contact, long distance data transmission, and simplified integration of synapses**. Because of this, one synaptic photomemristor can process multiple sensory inputs.
2. With respect to the functionality, our systems demonstrate **perceptual learning, memorizing and recognizing capabilities with dimensionality-reduced features**.
3. With respect to the coding algorithm, our systems demonstrate not only rate and temporal coding, but also **hardware-based feature coding with dimensionality reduction** that condenses raw data to be informative and non-redundant, facilitating subsequent processing.

In the Discussion, we now more clearly articulate the novelty of our research. A direct comparison with other artificial nerve systems is provide in Table S2 of the Supplementary Information.

Reviewer comment

For artificial afferent nerve system, the retention time or decay time of PSCs are important. The biological sensor system and previously reported artificial afferent nerve showed a short retention time because the sensor signal needs to be decayed immediately right after the elimination of stimulation for accurate sensing. The retention time can be seen in Fig S6 or 2d, which is relatively long for an artificial sensory nerve system. The reviewer thinks that previously remained postsynaptic current may still affect the following results. So, can the recognition process be operated continuously with different patterns without sufficient time intervals?

Our response

We agree with the reviewer that the retention time or decay time of PSCs could affect the next result in sensing or the recognition of patterns. However, in applications where the previous result needs to be memorized or combined with the next result, long retention times

are required. Therefore, both short and long decay times are important for neuromorphic sensing and processing.

To recognize different patterns continuously, two methods can be exploited in our spiking afferent nerve system:

1. Use of a **lower bias** to read the PSC of the synaptic photomemristors. As shown in Figure R1 below, the PSC read at a low bias of 0.75 V decays almost instantly after the pressure input is ended. At a bias of 1.25 V, the decay time is longer. The spiking rate in both cases is identical (60 Hz), signifying identical sensing capabilities at different biases (rate coding).

Fig. R1. PSCs read at a bias of 0.75 V and 1.25 V before, during, and after a pressure input.

2. Use of a **negative voltage pulse** to quickly reset the PSC to its initial state. We revised Figure S6 (also shown below) to demonstrate that a short negative voltage pulse across the synaptic photomemristor resets the PSC (see panel (d)), allowing the continuous execution of different tasks.

Fig. S6. (a) Current response of a synaptic photomemristor to a single light pulse. (b) Current change ratio in response to light pulses of different duration. The current change ratio decrease with decreasing pulse width. For a 1 ms light pulse, the current change ratio is about 13%. (c) Resistance switching cycles of the ITO/ZnO/Nb-STO optoelectronic memristor. (d) Demonstration that the persistent photoconductivity effect of the ITO/ZnO/Nb-STO is quickly reset by the application of a negative voltage pulse. In the experiment, the PSC is reset to its initial value by a negative voltage pulse with amplitude -3 V and duration 0.5 ms.

Reviewer #2

Reviewer comment

This manuscript is related on the emulation of tactile sensory coding and learning with artificial and bioinspired afferent nerves. The afferent nerves are made of a combination of devices including 2D material-based pressure sensors, ring oscillators with off-the-shelf electronics and photomemristors. Based on this system, functions related on tactile sensory and processing are demonstrated including processing of Morse code and braille alphabet. By having an array of pressure sensors and ring oscillators coupled with photomemristors, the authors also demonstrated dimensionality reduction of the data stream in hard-written letters. The manuscript is well organized and the results are presented in an easy to follow manner. However, the main argument of the manuscript is that in those systems a non-volatile end-device (memristor) is missing and this would enhance the functionality. I am not really sure about this argument for all demonstrations of the manuscript, since in the past Morse code and braille alphabet has been also demonstrated with volatile end-devices (Science 360, 6392, 998-1003 (2018) and Science Advances 4, 11, eaat7387 (2018)). Therefore, there is an overlap with the aforementioned works and Figures 1-3 of the manuscript. Nevertheless, the actual merit of the manuscript as well as the advantage of having non-volatile end-devices is the demonstration of dimensionality reduction in hard-written letters (Figure 4). To conclude, I believe that this work presents some interesting concepts in dimensionality reduction and could be published in Nat Commun., if the manuscript is more situated around the concept of Figure 4.

Our response

We would like to thank the reviewer for his/her positive assessment of our work, especially regarding the demonstration of dimensionality reduction in hard-written letters (Figure 4). Figures 1-3 and the corresponding text focus on the properties and functions of our spiking afferent nerve system. There, we demonstrate the implementation of rate coding, temporal coding, and a combination of these coding schemes. These bio-inspired coding methods are more robust than voltage amplitude coding, which is used often in other artificial afferent nerve systems. In the revised version of the manuscript, we introduce another coding scheme based on spike counting (new Figure S12 in the Supplementary Information). Moreover, we demonstrate the advantage of optical communication between discrete modules, allowing non-contact, long distance data transmission, and simplified integration of synapses. Because of this, one synaptic photomemristor can easily process multiple sensory inputs (Fig 3a-c).

Figure 4 shows how the integration of optoelectronic memristors enables hardware-based dimensionality-reduced feature extraction and learning with recognizing and memorizing capabilities. To emphasis this particular advantage of our system more, we extended the results on dimensionality reduction. In the original manuscript, we already demonstrated that our spiking afferent nerve can learn, recognize and memorize individual handwritten letters

of the alphabet using dimensionality-reduced feature codes. In the revised manuscript, we go beyond this result by showing that further reductions of the dimensionality are possible in the learning, recognition and memory of full words. The new experiments are summarized in Figure 5 (see below) and on page 11 of the revised manuscript the new results are described as follows: "Further dimensionality reduction in the classification of handwritten words is possible by combining the vectors of two subsequent letters. This attractive feature of our spiking afferent nerve system is demonstrated for the word 'APPLE' in Figure 5a and 5b. In this realization, the dimensionality of the word is reduced from twenty-five to fifteen, limiting the number of required photomemristors. Similarly, the handwritten words 'ORANGE', 'BANANA', 'PEAR', 'CHERRY', and 'GRAPE' can be represented also by fifteen-dimensional vectors (Figure S20). To classify these words, we built an artificial neural network (Figure 5c). The network consists of fifteen inputs and six outputs, corresponding to the elements of the fifteen-dimensional vectors and the six handwritten words, respectively. We trained the artificial neural network by repeated writing of the words. As shown in Figure 5d-5i, all words are recognized successfully after only four training cycles, as each output neuron responds to only one input word (Figure 5j). In this proof-of-principle experiment, the words are relatively short. We therefore combined only two alphabet letters. For the processing of longer words, three or more alphabet letters could be combined to reduce the vector dimensionality further and recognize words efficiently."

The new Figure S20 is also shown below. We thank the reviewer for suggesting an extension of the part on dimensionality reduction.

Fig. 5 | Classification of handwritten words with enhanced dimensionality reduction. a, Schematic diagram of the handwritten word 'APPLE'. **b**, Dimensionality reduction of the vector representing 'APPLE' from twenty-five (five per letter) to fifteen by combining 'A' and 'P' and 'P' and 'L'. **c**, Schematic diagram of the artificial neural network that processes the dimensionality-reduced vectors. The fifteen vector elements are used as inputs and the six output neurons represent the six handwritten words. **d-i**, Training results for the recognition of (d) 'APPLE', (e) 'ORANGE', (f) 'BANANA', (g) 'PEAR', (h) 'CHERRY', and (i) 'GRAPE' during the first 10 epochs. **j**, Response of the six output neurons upon handwriting of the fruit names after the neural network is trained.

Fig. S20. Color map of dimensionality-reduced features for the words ‘ORANGE’, ‘BANANA’, ‘PEAR’, ‘CHERRY’, and ‘GRAPE’. The corresponding color map for ‘APPLE’ is shown in Figure 5b of the main manuscript. Reduction of the vector dimensionality is achieved by combining the pressure inputs from two subsequent handwritten letters (i.e. use of one synaptic photomemristor instead of two). Depending on the word, this reduces the dimensionality from twenty, twenty-five, or thirty to fifteen.

Reviewer #3

Reviewer comment

While the author have developed an interesting intelligent mechanotransduction system, I don't see that this will be of general enough interest for Nature Communications, given the work of people like Gordon Cheng (A Comprehensive Realization of Robot Skin) or Zhu (Bilayered Oxide-Based Cognitive Memristor with Brain-Inspired Learning Activities). It seems more appropriate for a material science or IEEE journal.

My concern for the most part isn't technical (the MXene-based sensor and photomemristor seems reasonable and technically well-conceived, with clever hardware-based processing; I also have no problems with the analysis used). It is more a matter of scope for a generalist journal. That said, I do wonder about the form factor of the different versions of the device?

Our response

We would like to thank the reviewer for taking his/her precious time to read and comment on our work. We do not share the reviewer's concern about the general interest of our work for the following reasons:

Humans use tactile sensory coding and learning in biological somatosensory systems to interact with the world. Emulation of tactile sensory detection, coding, processing and learning in bio-inspired distributed and hierarchical architectures provides a promising strategy for in-memory computing of sensory information, biomimetic sensing, neurorobotics, and human-machine interactions. Recently, with a biorealistic, efficient, and robust neuromorphic architecture, Y. Kim and coworkers reported a spiking afferent nerve that combines resistive pressure sensors, ring oscillators, and a synaptic transistor to detect, convert and integrate pressure information (Science 360, 998 (2018)). While the degree of biological emulation in terms of distributed networks in this paper is a milestone, the hardware does not exhibit plasticity, which is essential for the learning and memorizing of tactile information. Biological somatosensory systems do not only detect pressure information but, very importantly, also learn from and memorize multiple (and often simultaneous) inputs through complex neural coding principles. These functionalities allow biological systems to deal with large amount of sensory data. Learning and memorizing capabilities also enable humans to learn from and adapt to their environment. While it is broadly recognized that the implementation of these functions at the hardware level of artificial afferent nerve systems is essential for robotics and neurorobotics, it has not been realized yet.

In our manuscript, we report an optoelectronic spiking afferent nerve system with tactile information sensing, optical spike coding, learning and memorizing capabilities. In our system, 2D MXene-based pressure sensors detect the tactile information, an ADC-LED circuit converts the DC signal to digital optical spikes in a hierarchical architecture, and photomemristors

integrate the optical spikes with encoded pressure information. Using rate and temporal coding algorithms (the two major coding principles in biological systems) or a combination of both, we demonstrate pressure information recognition with supervised learning. Moreover, automatic feature extraction (coding) and learning in a dimensionality-reduced architecture of multiple spiking nerves is shown. We use this capability to recognize, learn, and memorize handwritten letters of the alphabet and entire words. The biomimetic learning and memorizing capabilities achieved through the inclusion of plasticity at the hardware level and the utilization of complex coding principles provide a promising strategy for smart sensing, neurorobotics and human-machine interactions.

Above all, our work on bioinspired optoelectronic spiking afferent nerves combines multidisciplinary research on materials, device integration, and computational algorithms, especially 2D materials, optical communications, photomemristive devices, e-skin, sensory spiking neural networks, sensory coding, feature engineering, dimensionality reduction, perceptual learning, brain-inspired computing, neurorobotics, and artificial intelligence. We therefore believe that our work will be of interest to a broad and diverse readership and attract the attention of researchers from various scientific disciplines.

REVIEWERS' COMMENTS:

Reviewer #1 (Remarks to the Author):

In the manuscript titled "Tactile sensory coding and learning with bioinspired optoelectronic spiking afferent nerves", the authors reported the tactile sensor system with an optoelectronic artificial nerve. The optoelectronic sensory system has been made of MXene pressure sensor, LED-coupled with the ADC circuit which includes ring oscillator and edge detector, which converts electronic signals to optical spikes. The authors demonstrated the tactile sensing system with recognition of Morse code, 5-letter-word, braille character and also dimensionality-reduced feature. The concerns of the reviewers were adequately addressed. So the manuscript is strengthened and seems more clear after proper revision works. Therefore, the manuscript can be accepted by nature communication now.

Reviewer #2 (Remarks to the Author):

In the revised manuscript, the authors addressed most of my questions. The manuscript is now more focused on dimensionality reduction aspects of Fig. 4 and Fig. 5. I believe that now the manuscript presents more clearly its unique parts and therefore it deserves publication in Nat. Commun.

Reviewer #1

Reviewer comment

In the manuscript titled “Tactile sensory coding and learning with bioinspired optoelectronic spiking afferent nerves”, the authors reported the tactile sensor system with an optoelectronic artificial nerve. The optoelectronic sensory system has been made of MXene pressure sensor, LED-coupled with the ADC circuit which includes ring oscillator and edge detector, which converts electronic signals to optical spikes. The authors demonstrated the tactile sensing system with recognition of Morse code, 5-letter-word, braille character and also dimensionality-reduced feature. The concerns of the reviewers were adequately addressed. So the manuscript is strengthened and seems more clear after proper revision works. Therefore, the manuscript can be accepted by nature communication now.

Our response

We would like to thank the reviewer for the positive assessment of our work and his/her recommendation to publish our manuscript in Nature Communications.

Reviewer #2

Reviewer comment

In the revised manuscript, the authors addressed most of my questions. The manuscript is now more focused on dimensionality reduction aspects of Fig. 4 and Fig. 5. I believe that now the manuscript presents more clearly its unique parts and therefore it deserves publication in Nat. Commun.

Our response

We would like to thank the reviewer for the positive assessment of our work and his/her recommendation to publish our manuscript in Nature Communications.